# One-Year Analysis of Autologous Peripheral Blood Mononuclear Cells as Adjuvant Therapy in Treatment of Diabetic Revascularizable Patients Affected by Chronic Limb-Threatening Ischemia: Real-World Data from Italian Registry ROTARI

**DOI:** 10.3390/jcm13175275

**Published:** 2024-09-05

**Authors:** Sergio Furgiuele, Enrico Cappello, Massimo Ruggeri, Daniele Camilli, Giancarlo Palasciano, Massimiliano Walter Guerrieri, Stefano Michelagnoli, Vittorio Dorrucci, Francesco Pompeo

**Affiliations:** 1Unit of Vascular and Endovascular Surgery, High Specialty Hospital “Mediterranea”, 80122 Napoli, Italy; 2Second Unit of Vascular and Endovascular Surgery, IRCCS Neuromed, 86077 Pozzilli, Italy; enrico.cappello@yahoo.it (E.C.); pompeo@neuromed.it (F.P.); 3Unit of Vascular Surgery, San Camillo de Lellis Hospital, 02100 Rieti, Italy; m.ruggeri@asl.rieti.it; 4Casa di Cura Santa Caterina della Rosa Asl RM 2, 00176 Roma, Italy; daniele.camilli@asl.roma2.it; 5Vascular Surgery Unit, Department of Medicine, Surgery and Neuroscience, University of Siena, 53100 Siena, Italy; giancarlo.palasciano@unisi.it (G.P.); wguerrieri63@gmail.com (M.W.G.); 6UOC Vascular Surgery, San Donato Hospital, 52100 Arezzo, Italy; 7Vascular and Endovascular Surgery Unit, San Giovanni di Dio Hospital, 50143 Florence, Italy; stefano.michelagnoli@uslcentro.toscana.it; 8Department of Vascular Surgery, Umberto I Hospital, 96100 Venice, Italy; vdorrucci@ulss12.ve.it

**Keywords:** CLTI, PBMNC, adjuvant therapy, amputation, wound healing, diabetes

## Abstract

Wounds in diabetic patients with peripheral arterial disease (PAD) may be poorly responsive to revascularization and conventional therapies. **Background/Objective**: This study’s objective is to analyze the results of regenerative cell therapy with peripheral blood mononuclear cells (PBMNCs) as an adjuvant to revascularization. **Methods:** This study is based on 168 patients treated with endovascular revascularization below the knee plus three PBMNC implants. The follow-up included clinical outcomes at 1-2-3-6 and 12 months based on amputations, wound healing, pain, and TcPO2. **Results**: The results at 1 year for 122 cases showed a limb rescue rate of 94.26%, a complete wound healing in 65.59% of patients, and an improvement in the wound area, significant pain relief, and increased peripheral oxygenation. In total, 64.51% of patients completely healed at 6 months, compared to the longer wound healing time reported in the literature in the same cohort of patients, suggesting that PBMNCs have an adjuvant effect in wound healing after revascularization. **Conclusions**: PBMNC regenerative therapy is a safe and promising treatment for diabetic PAD. In line with previous experiences, this registry shows improved healing in diabetic patients with below-the-knee arteriopathy. The findings support the use of this cell therapy and advocate for further research.

## 1. Introduction

Chronic limb-threatening ischemia (CLTI) is the end stage of peripheral artery disease (PAD), characterized by significant mortality, limb loss, pain, and diminished quality of life [1]. The risk for the development of CLTI in patients with peripheral arterial disease (PAD) was observed to be higher in men, in patients who suffered of stroke or heart failure, and in patients with diabetes mellitus (DM) [2]. A higher risk of amputation was reported in CLTI patients, also after a successful revascularization, when patients showed a greatest tissue loss [3].

In an extensive analysis, retrospectively obtained from a cohort of 41,882 patients hospitalized due to PAD, of which 49% suffered from CLTI, the rates of amputation at 4 years were 35.3%, and 67.3% for Rutherford class 5 and 6, respectively [4]. Moreover, diabetic PAD patients show the highest risk of developing CLTI, with 25% 1-year mortality, which reaches 45% after limb amputation and a majority prevalence of gangrene [1]. Revascularization is indicated to treat disabling claudication and/or rest pain and trophic lesion with foot transcutaneous oximetry (TcPO2) < 30 mmHg, or any sign of healing after one month [5]. DM patients often showed both plaque-based, atherosclerotic Big Artery Disease (BAD) together with the presence of Small Artery Disease (SAD), obstructing the below-the-ankle arteries [6]. SAD has been related with poor outcomes in terms of wound healing, time to healing, limb salvage, and survival, suggesting that SAD is an untreatable disease, leading to the failure of the foot vascular network, and jeopardizing the fate of the limb, despite any successful revascularization of BAD [7,8].

Since the discovery that immune cells can contribute to angiogenesis and arteriogenesis [9,10,11,12,13,14,15], there has been an increasing number of studies testing the efficacy of autologous cell therapies for the treatment of non-revascularizable CLTI patients to obtain the formation of collaterals leading to increased blood flow and tissue regeneration in non-healing trophic lesions [16,17,18,19,20,21]. Peripheral blood mononuclear cells, PBMNCs, which consist of a heterogeneous population of lymphocytes and monocytes, CD34+ hematopoietic stem cells, and Endothelial Progenitor Cells, EPCs, seem to be the most promising autologous cell therapy for the treatment of no-option CLTI patients [22]. This is probably due to the dual mechanisms of action: the angiogenic and arteriogenic potency of PBMNCs [13,23,24,25,26] together with their ability to induce tissue regeneration, especially in chronic wounds, switching the M1 inflammatory macrophages phenotype in the M2 regenerative one [27,28]. PBMNCs have also shown several interesting features such as ease of collection, which allow multiple repeated implants, given that the frequency of implants seems to work better than a single implant with a higher dose [29,30]. Moreover, while CD34+ stem cells from diabetic patients do not respond to hypoxia and show a reduced angiogenic effect [31,32], both the monocyte and lymphocyte population of the diabetic patient remain less impaired, keeping the ability to stimulate angiogenesis and the healing of diabetic ulcers [32]. In line with these in vitro data, two randomized trials have demonstrated the non-inferiority of PBMNCs compared to CD34+ purified stem cells from peripheral blood and to bone marrow mononuclear BM-MNCs [33,34]. The meta-analysis by Rigato et al. [17] conducted on 19 randomized controlled trials (RCTs), 7 non-randomized trials, and 41 non-controlled trials for a total of 1177 patients showed that cell therapy reduced the risk of amputation by 37%, improved amputation-free survival by 18%, and improved wound healing by 59%, even if it was not able to affect mortality. Cell therapy also significantly increased transcutaneous oxygen tension and reduced rest pain [17]. Moreover, PBMNCs perform better than BM-MNCs and mesenchymal stem cells from bone marrow BM-MSCs, significantly reducing amputations [17]. Remarkably, the amputation rate improved more in trials with a high prevalence of diabetes mellitus, and no severe adverse events have been associated with the treatment [17]. Accordingly, a previous meta-analysis of 16 RCTs and 774 patients showed that PBMNCs reduce major amputations and significantly increase wound healing [35]. PBMNC therapy can also be considered adjuvant therapy in patients undergoing revascularization [36,37] and in diabetic patients where revascularization alone could be insufficient for efficient wound healing [6,7,38]. Persiani et al. published, for the first time, preliminary data on PBMNC implant as adjuvant therapy in revascularization on 32 diabetic patients with ulcerative lesions [37].

Based on these encouraging data, under the patronage of the Italian College of Vascular Chiefs (Collegio Italiano dei Primari Ospedalieri di Chirurgia Vascolare-www.collprimvasc.org), the ROTARI (Italian Regenerative Adjuvant Therapy Observational Register) was set-up (1 March 2017).

The ROTARI is a national register used to collect data (major and minor amputations, wound healing, pain, and TcPO2) in CLTI diabetic patients with ulcerative lesions that undergo below-the-knee (BTK) endovascular revascularization and PBMNCs as adjuvant therapy to improve wound healing.

## 2. Materials and Methods

The ROTARI is a multicenter, national, prospective real-word registry evaluating autologous PBMNCs as adjuvant therapy for CLTI diabetic patients with trophic lesions revascularized BTK. From March 2017 to February 2020, a total of 168 patients underwent BTK endovascular revascularization and three cycles of PBMNCs at 35–40 days each (Table 1 and Table 2). Informed consent was obtained from all subjects involved in this study.

The inclusion criteria were the diabetic patients’ candidate to BTK revascularization with ulcers (Rutheford 5).

Exclusion criteria were moderate or severe infection; anemia (Hb < 8 g/dL); coagulation disorder/thrombocytopenia (PLT < 50,000/µL); or active cancer/leukemia or lymphoma hematological disease.

The follow-up at one year was possible only in 122 cases out of 168 due to COVID-19, which prevented the participants from attending the reassessment. Therefore, the successive data exclusively concern the 122 patients who completed the one-year follow-up.

All patients had ulcerative lesions. Staging at T0 was performed with the Wagner scale (Table 3).

The primary endpoints were limb salvage and wound healing according to the Wagner scale. Wound healing was assessed as the patients completely healed, time to healing (days from revascularization), Wagner scale wound evolution (Grade 1 signifies a superficial ulcer on the outer layers of skin, Grade 2 is a deep ulcer, Grade 3 is an ulcer with bone involvement, Grade 4 means that there is gangrene or dead tissue in the front of the foot, and Grade 5 means that the gangrene has spread to the entire foot), and the wound area (cm^2^).

The secondary endpoints were the evaluation of pain (VAS scale) and tissue peripheral oxygenation (TCpO2). Data were collected in the ROTARI database and analyzed retrospectively. Endovascular revascularization consisted in BTK percutaneous transluminal angioplasty of below-the-knee arteries. Revascularization details are reported in Table 2. The endovascular revascularization was performed under local anesthesia, and the homolateral percutaneous femoral access was cannulated.

### PBMNC Concentration

Autologous PBMNC concentrate was produced by MonoCells Solution/Pall Celeris-/Hematrate (Athena Cells Therapies Technologies, Brindisi, Italy/Pall Medical, Port Washington, NY, USA/Cook Regentec, Indianapolis, IN, USA), a filtration-based point-of-care device for the rapid preparation of PBMNCs from 120 mL of anticoagulated blood, for use in human cell therapy applications. The cell product obtained was extensively characterized in terms of composition, recovery, and FACS cell population analysis [34]. All procedures were performed in an operating room with anesthesiologic support (propofol and/or peripheral block). After the appropriate surgical debridement of the wound bed, multiple perilesional and intramuscular injections of 8-10 mL PBMNC cell suspensions (0.25 mL in boluses) were immediately injected along the relevant axis below the knee, at intervals of 1–2 cm and to a mean depth of 1.5–2 cm, using a 21 G needle. PBMNCs were implanted the first time together with the revascularization. The other two PBMNC implants were performed for each patient at intervals of 30–45 days from each other.

Follow-up was performed at 1-2-3-6 and 12 months after revascularization (T0).

Descriptive statistics included proportions for categorical and mean (standard deviation (SD) or median (interquartile range) for continuous variables, according to the skewness of the data distribution. No assumptions were made for missing data.

## 3. Results

### 3.1. PBMNC Implants

PBMNCs were enriched four-fold in comparison to peripheral blood. The average dose of PBMNCs implanted was 1.06 ± 0.28 × 10^8^ (0.9 ± 0.25 × 10^8^ limphocytes and 0.16 ± 0.04 × 10^8^ monocytes. The CD34+ progenitor cell subpopulation was also enriched by 5.6% ± 4.2% with a mean cell count of 1.37 × 10^6^. No minor or major adverse events related to autologous PBMNC implants were observed.

### 3.2. Limb Salvage

In our cohort, the rate of limb salvage was 94.26% at 1 year. Major and minor amputation are reported in Table 4 and Table 5.

#### 3.2.1. Major Amputation (Transfemoral or Transtibial)

Only 7 out of 122 patients (5.74%) underwent major amputation. A progressive decrease in amputations was observed from the first to the third month with four amputations within 30 days from T0, two within 60 days, and one within 90 days (Table 4). No major amputations were recorded after the third month. Major amputations were performed only in patients with Wagner scale G4 (4 out of 24, equal to 16.66%) and G5 (3 out of 3, 100%) (Table 5).

#### 3.2.2. Minor Amputation (Digital and Ray Amputation of the Toe, Trans Metatarsal Amputation, and Lisfranc and Chopart Amputation)

A total of 22 patients out of 122 (18.03%) underwent minor amputation. Compared to major amputations, minor amputations are inversely proportional to the time elapsed since T0. In fact, 13 minor amputations were performed within the first month, another 2 within the second month, 4 within the third month, and 3 within the sixth month. No patients underwent minor amputation after 6 months (Table 4). Minor amputations were performed in each stage of the Wagner scale but in different percentages, excluding major amputations (Table 5): in G1, 6 cases out of 58 (10.34%); in G2, 4 out of 29 (13.79%); in G3, 2 out of 8 (25%); and in G4, 10 out of 24 (41.66%).

### 3.3. Wound Healing

Wound healing was assessed in patients with no major or minor amputations (93 cases) in order to evaluate the wound evolution and the time to healing. In this cohort, complete wound healing was obtained in 61 cases out of 93 at 12 months (65.59%) (Table 6)

#### Wound Healing Evolution

The wound healing evolution is shown in Figure 1. Improvements in deeper lesions (G3–G4) and the healing of superficial lesions (G1–G2) were observed. At 1 year, we observed completely healed patients (G0 = 61 cases) and superficial lesions (G1 = 32 cases) only, for any wound at G2, G3, or G4.

In the 32 patients who were not healed at 1 year, we observed a significant reduction in the lesion area, as reported in Figure 2.

### 3.4. Pain

Pain on the VAS scale was considered only in patients that did not undergo amputations (major or minor) (n = 93). A rapid improvement in pain was observed at 1 month, with progressive improvement up to 12 months, as reported in Figure 3.

### 3.5. TcPO2

Tissue peripheral oxygenation was considered only in patients that did not undergo amputation (major or minor) (n = 93). After a rapid increase at 1 month, we observed a subsequent increase up to the third month, followed by a slight decrease at 6 months and substantial stabilization at 12 months (Figure 4).

## 4. Discussion

While arterial revascularization is often considered the guarantee for clinical improvement, this is not always true in diabetic patients, or in the absence of neuropathy and infections [7,8,39,40]. This is due to an extended and dysregulated cellular and molecular healing mechanism in diabetic patients that is attributed to the inadequate coordination of blood vessel formation, disturbed balance in angiogenic and growth factors, abnormally prolonged inflammation, and an impaired immune response [41,42,43,44].

As PBMNCs have been shown to be the most promising therapy for no-option CLTI patients [17,22], and more specifically in diabetic patients [18,19,20,21], we hypothesized that PBMNCs could be used as adjuvant therapy, in terms of wound healing in diabetic patients, after or together with revascularizing procedures [37].

These data are supported by biological and cellular PBMNC mechanisms of action [45,46]; in fact, monocytes and lymphocytes play a key role in both angiogenesis/arteriogenesis [13,23,24,25,26], and in tissue regeneration, especially in diabetic patients [28]. The observed impaired transition of diabetic wound macrophages from pro-inflammatory macrophage M1 phenotypes to anti-inflammatory pro-regenerative M2 phenotypes might represent a key issue for delayed diabetic ulcer healing, even after successful revascularization [45,46]. It has been observed in biopsies of diabetic non-healing ulcers after revascularization that PBMNCs implanted in the perilesional area polarize M1 macrophages in M2 and promote relevant changes in the molecular setting over time to increase wound healing [27].

A retrospective trial on 367 CLTI diabetic patients showed that despite the disappearance of the ischemic component due to efficient revascularization, the observed wound healing of the foot was extremely slow, requiring up to 2 years [38]. In our cohort of diabetic patients, the healing was obtained in 64.51% at 6 months and in 65.59% at one year.

Caetano et al. [47] published, in 2020, a single-center retrospective study including 314 patients with DFUs, of which 285 who underwent successful endovascular revascularization reported a major amputation rate of 3.9%, a minor amputation rate of 52%, and complete wound healing of 53.7% at 1-year follow-up. Comparing this with our data, we observed a higher rate of major amputation of 5.74% vs. 3.9% probably because 22% of this cohort of patients showed only above-the-knee arteriopathy. Despite these data, we observed a significatively lower minor amputation rate (18% vs. 52%) compared to that of Caetano et al. These data are in line with the higher wound healing rate at one year (65.59% vs. 53.7%) probably due to PBMNC implants.

In 2016, a meta-analysis on 56 studies by the International Working Group on the Diabetic Foot (IWGDF) showed a major amputation rate after successful revascularization of 5% with complete DFU healing from 21% to 54% [48]. The same group published, in 2020, a major amputation rate of 10% at 1 year and a wound healing median rate of 60% at 1 year [49].

Ferraresi et al. [8] observed, in a similar cohort of patients with tissue loss, only 45.5% complete healing after a successful revascularization rate at follow-ups of 19 ± 10.8 months.

Regarding the pain and TcPO2 evaluation, these data seem to confirm what was previously published in a monocentric study on patients treated with autologous PBMNCs as adjunctive therapy to revascularization [37]. Even in the presence of revascularization, PBMNC therapy could have an improving role both in pain and peripheral tissue oxygenation, related to specific biological characteristics and mechanisms of action. Regarding the pain relief mechanism, PBMNCs and, in particular, monocyte/macrophages, may play a significant role in relation to the secretion of natural opioids released by M2 macrophages [50,51,52,53]. Interestingly, a new mechanism of action has been recently described related to the analgesic effect, based on mitochondrial transfer to sensory neurons to resolve inflammatory pain [54].

Regarding peripheral tissue oxygenation, the increase in TcPO2 after PBMNC implant was previously observed in diabetic no-option CLTI patients [18,20,21,55,56,57], because of their extensively described angiogenesis potency [11,13,23,58]. Moreover, Panunzi et al. [18] observed a correlation between PBMNC implants and increased microvascular density responsible for perfusion in no-option diabetic CLTI patients. The authors observed that the TcPO2 improvement was more remarkable when PBMNC therapy was performed three times, as in the ROTARI register. Moreover, patients that showed higher levels of TcPO2 (≥40 mmHg) were characterized by a higher concentration of extracellular vesicles (EVs) with a smaller-sized end (30–100 nm EVs) in the plasma with respect to patients with TcPO2 ≤ 40 mmHg [18]. Interestingly, post-PBMNC implant angiographies were performed in four patients, and an increase in collateral vessels indicating vascular remodeling was observed [18].

Our data highlight that patients with CLTI and advanced foot gangrene (G5) all underwent major amputation. Furthermore, we noticed also that all major amputations (3 out of 3; 100% G5 and 4 out of 24; 16.6% of G4) occurred within the first three months. These data, which are comparable to those of another study [38], identify the ineffectiveness of cellular therapy in the most severe cases. Conversely, in our cohort of PBMNC-treated patients, the minor amputation rate appears lower than in diabetic patients revascularized only [47], occurring, similarly, to major amputations, in 86% of cases within the first three months, and in no cases after 6 months. These data suggest the possible role of PBMNC therapy in improving angiogenesis and wound healing and capability of reducing the number of demolitive interventions.

ROTARI data show not only a positive result in the wound healing result at 12 months but also suggest a faster improvement and healing of the wound during the follow-up. In fact, our data show, already at 6 months, a substantially overlapping wound healing rate with the one observed at 12 months. This wound healing rate, with 64.51% of patients completely healed at 6 months, is considerably shorter in comparison to the wound healing reported in the literature [8,38,40,47,48,49], suggesting that PBMNCs have an adjuvant effect in wound healing. Regarding the pain symptomatology in our patient cohort, we believe that the initial improvement, which is certainly correlated with revascularization, is supported and consolidated in the following months due to the improvement in wound healing promoted also by PBMNCs.

Finally, concerning peripheral oxygenation, it is significant to note that the instantaneous increase observed in the first month, which is certainly related to endovascular BTK revascularization, presents a subsequent significant increase favored by the repeated implantation of PBMNCs on an angiogenic basis, as previously observed [18].

A major limitation of this study resides in its uncontrolled nature. Moreover, the available literature is rarely comparable due to the characteristics of the patient cohort: most studies include non-diabetic patients, patients without trophic lesion, without revascularization or after revascularization failure, or revascularized but not specifically patients with BTK arteriopathy.

In the absence of a comparator group, the results of this study cannot support a direct comparison with BTK revascularization, only versus revascularization with adjuvant PBMNC implant. On the other hand, the evaluation of PBMNC effectiveness was related to the clinical outcomes in a full real-world clinical cohort treated at different centers. No less important is the total absence of side effects of PBMNC implants.

## 5. Conclusions

The challenges of developing treatments for this highly critical diabetic patient group necessitate a combination or sequential innovative treatment approach.

The data of the ROTARI multicenter registry, although lacking a control group, suggest the potential of an autologous regenerative cellular therapy with PBMNCs as an adjunct to distal below-knee revascularizations in diabetic patients, with no adverse or related side effects. The biological rationale of this therapy and the clinical outcome obtained in these extremely critical patients, both for distal arterial pathology and concomitant diabetic pathology, encourage further research in this direction.

## Figures and Tables

**Figure 1 jcm-13-05275-f001:**
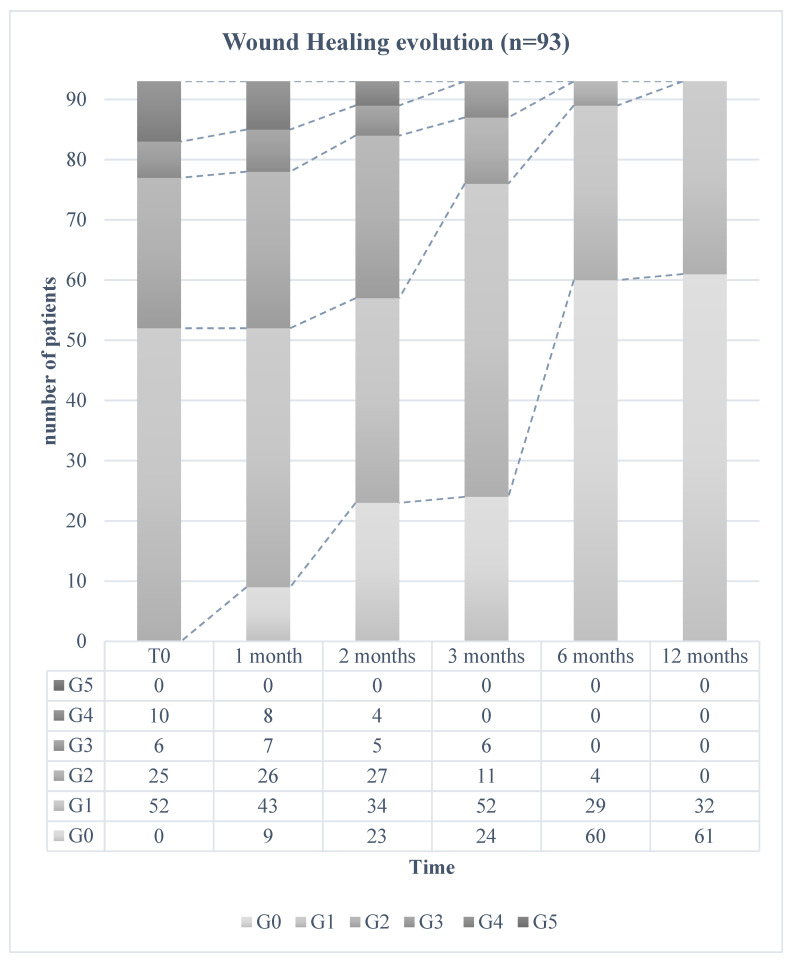
Wound healing evolution (n = 93).

**Figure 2 jcm-13-05275-f002:**
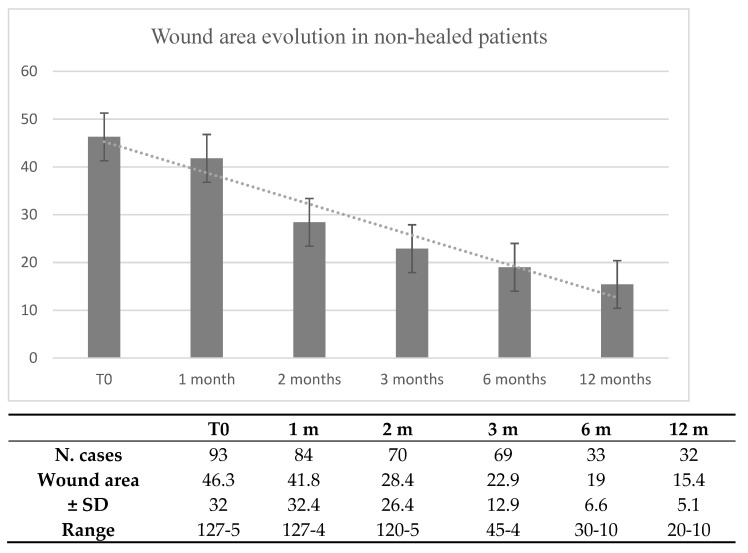
Wound area evolution in non-healed patients.

**Figure 3 jcm-13-05275-f003:**
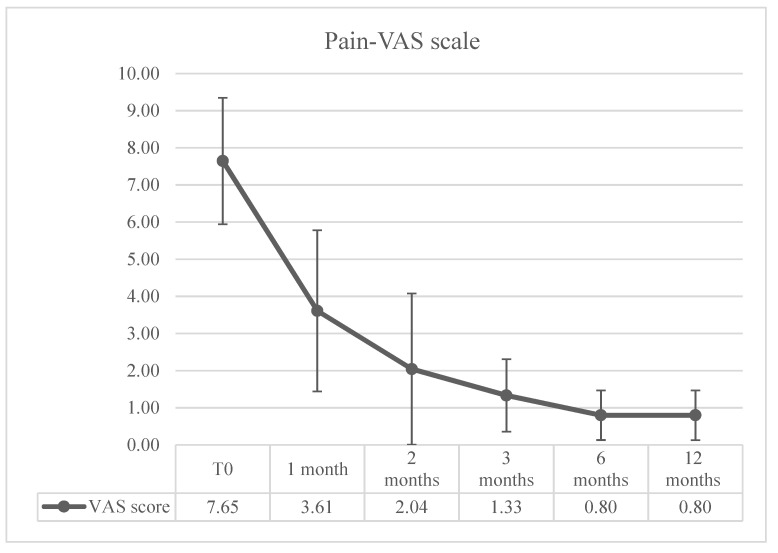
Pain evaluation–Pain VAS Scale (n = 93).

**Figure 4 jcm-13-05275-f004:**
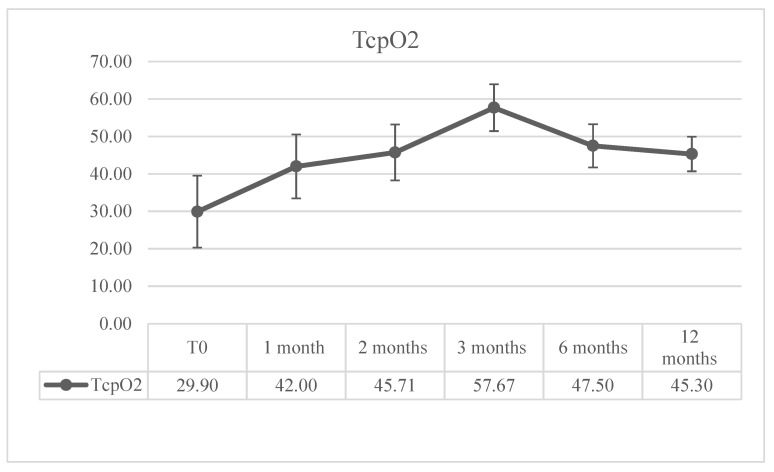
TCPO2 (n = 93).

**Table 1 jcm-13-05275-t001:** Main characteristics of the enrolled diabetic patient cohort.

n = 122	Characteristic n (%)
Mean age	74.1 ± 11.8
Gender	82 M, 40 F
Hypertension	95 (77.8%)
Dyslipidemia	80 (65.5%)
Cardiopathy	38 (31.2%)
Smokers/ex-smokers	70 (57.4%)
Renal failure	14 (11.5%)

Data are presented as n (%) or mean ± SD unless otherwise indicated.

**Table 2 jcm-13-05275-t002:** PTA revascularization.

n = 122	n Revascularized Vessel
Patients revascularized	
1 vessel	83
2 vessels	22
3 vessels	17
Popliteal	67
Anterior Tibial Artery	38
Peroneal Tibial Artery	14
Peroneal Artery	28
Posterior Tibial artery	31
DCB *	30
Stenting	4

* DCB—drug-coated balloon.

**Table 3 jcm-13-05275-t003:** Stage of ulcerative lesions—Wagner scale at T0.

Stage	n Cases (%)
G0	0
G1	58 (47.5%)
G2	29 (23.8%)
G3	8 (6.5%)
G4	24 (19.6%)
G5	3 (2.4%)

Cases 122 patients who completed 1-year follow-up.

**Table 4 jcm-13-05275-t004:** Amputations stratified for time n = 122.

Time	Major Amp	Minor Amp
1 month	4	13
2 months	6	15
3 months	7	19
6 months	7	22
12 months	7	22

In total, 122 patients completed the follow-up at one year.

**Table 5 jcm-13-05275-t005:** Amputations stratified for Wagner scale at T0 n = 122.

Stage *	Major Amp	Minor Amp
G1	0	6
G2	0	4
G3	0	2
G4	4	10
G5	3	0

In total, 122 patients completed the follow-up at one year.* Wagner Scale

**Table 6 jcm-13-05275-t006:** Patients completely healed (n = 93).

Time (months)	n Healed Pts	%
0	0	0
1	9	9.67
2	23	24.73
3	24	25.81
6	60	64.51
12	61	65.59

## Data Availability

Data presented in this study are available on request from the corresponding author due to privacy and legal restrictions.

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
