# Peer review of "One-Year Analysis of Autologous Peripheral Blood Mononuclear Cells as Adjuvant Therapy in Treatment of Diabetic Revascularizable Patients Affected by Chronic Limb-Threatening Ischemia: Real-World Data from Italian Registry ROTARI"

_jcm, 2024, doi:10.3390/jcm13175275_

Round 1

Reviewer 1 Report

Comments and Suggestions for Authors

The authors present an interesting study examining the efficacy of regenerative approaches for ischaemic injury that manifests in diabetic wounds. Briefly, the authors recruited several patients who were in receipt of the therapy and measured baseline indices before doing a one-year follow up to determine the efficacy of the treatment for promoting revascularisation. While COVID-19 prevented all recruits returning for the follow up, a sizeable number did return, within which it was evident that the treatment did improve clinical outcomes. Overall this was a comprehensive and interesting study.

In reviewing the manuscript I made a couple of observations, the following should be considered when preparing a suitable resubmission.

1.      The writing for the most part is of a good standard, however, in the introduction for example more section breaks would benefit the flow of the piece. The authors should consider breaking the text up somewhat using section breaks in the introduction.

2.      When the authors suggests that COVID-19 was a barrier for completing the follow-up, could they clarify whether this prevented the participants from attending the reassessment, or was ‘contracting COVID-19’ an exclusion criteria or other?

3.      It would be preferred if the authors used Table 1 and Table 2 etc as their reference points rather than Tab1 and Tab2 or Tab.4 and Tab.5 as is used within the text.

Comments on the Quality of English Language

Those points pertaining to the writing of the manuscript can be found in the main body of my report. 

Author Response

Comment 1. The writing for the most part is of a good standard, however, in the introduction for example more section breaks would benefit the flow of the piece. The authors should consider breaking the text up somewhat using section breaks in the introduction. 

Replay 1. Thank you for the suggestion. We added some section breaks as suggested 

Comment 2.  When the authors suggest that COVID-19 was a barrier to completing the follow-up, could they clarify whether this prevented the participants from attending the reassessment, or was ‘contracting COVID-19’ an exclusion criteria or other?

Replay2:  COVID-19 due to critical access to hospitals was a barrier to attending reassessment. We clarify this in the material and method section (text added in red) specifying the issue. 

Comment 3 . It would be preferred if the authors used Table 1 and Table 2 etc as their reference points rather than Tab1 and Tab2 or Tab.4 and Tab.5 as is used within the text.

Replay 3. Corrected as per suggestion.

Reviewer 2 Report

Comments and Suggestions for Authors

Dear colleagues!
After review of the manuscript by Furgiuele et al. I have the following comments regarding its merit and scientific soundness. Overall, the communication reports a significant clinical observation with a large and interesting study power. The work would benefit from addition of statistical significance evaluations - at the moment only descriptive statistical data has been made available. 

Furthermore, in Figure 2 grayscale color legends are hard to understand - probably, coloured image would be a better choice.

One would also question the general possibility for comparison with other registers - e.g. revascularized and no treated by cell therapy. As far as Authors fairly noticed that no control group has been enrolled and used in the study reported: would that be a reasonable option to provide significant power to further works in this field?

Comments on the Quality of English Language

Generally, the manuscript is well-proofed and has minimal imperfections in terms of English language quality.

Author Response

Comments 1

The work would benefit from addition of statistical significance evaluations; at the moment only descriptive statistical data has been made available. 

Replay 1: Regretfully, descriptive analysis was the only approach available for statistical data analysis. 

Comments 2: Furthermore, in Figure 2, grayscale color legends are hard to understand—probably, a colored image would be a better choice.

Replay 2 corrected in gray scale from white to black. We also insert a bigger graph, which is easy to be read. Thank you for the suggestion.

Comments 3 One would also question the general possibility for comparison with other registers, e.g. revascularized and not treated by cell therapy. As far as Authors fairly noticed that no control group has been enrolled and used in the study reported: would that be a reasonable option to provide significant power to further works in this field?

In discussion were reported results from other papers reporting CLTI  revascularized  diabetic patients clinical outcome, while in the future we are planning to restart a new register that includes the control group.